# Factors Influencing Post-Marriage Education and Employment among Bangladeshi Women: A Cross-Sectional Analysis

**Bishwajit Ghose** [1,*] , **Iftekharul Haque** [2] and **Abdullah Al Mamun** [2]

1   Center for Social Capital and Environmental Research, Dhaka 1100, Bangladesh
2   School of International Development and Global Studies, University of Ottawa,
    Ottawa, ON K1N 6N5, Canada; ahaqu057@uottawa.ca (I.H.)
*   Correspondence: bghose@cscer.ca

**Abstract:** Higher education and employment are two key components of women's empowerment. However, many women fail to continue their studies or work after marriage, which can significantly reduce their empowerment potential, especially in countries with stark gender inequality such as in Bangladesh. In this study, our objective was to explore the individual, household and community factors associated with post-marriage education and employment among Bangladeshi women using data from the latest Bangladesh Demographic and Health Survey (BDHS 2017–18). Data were analysed using multivariate logistic regression methods. The results of the study show that a large proportion of the participants did not continue their studies (42.1%) or work (72.5%) after marriage, while only 3% of the participants studied and about 29.0% worked for more than 5 years after marriage. The most important factors associated with continuing to study after marriage include having access to a mobile phone (OR = 1.89, 95% CI = 1.62, 2.19), the husband's number of years of education (OR = 1.11, 95% CI = 1.08, 1.15), a higher household wealth index (OR = 1.27–4.31) and improved toilet facilities (OR = 1.36, 1.12, 1.65). Conversely, the number of children (OR = 0.69, 95% CI = 0.65, 0.73), living in rural areas (OR = 0.78, 95% CI = 0.68, 0.88) and residing in certain divisions are negatively associated with continuing to study after marriage. Women with a mobile phone (OR = 1.47, 95% CI = 1.06, 2.03) are more likely to continue working after marriage, while those with larger spousal age differences (OR = 0.33, 95% CI = 0.19, 0.58) and those living in the Chittagong division (OR = 0.53, 95% CI = 0.30, 0.96) are less likely to do so. The study indicates that a large proportion of Bangladeshi women do not continue their education or work after marriage. These findings underscore the significance of empowering women and addressing sociodemographic issues to promote education and work opportunities after marriage.

**Keywords:** education; employment; women's empowerment; Bangladesh





## 1. Introduction

Women's empowerment is regarded as a policy development priority worldwide given its critical importance in achieving the Sustainable Development Goals [1,2]. In general terms, women's empowerment refers to the process by which women gain power, agency and control over their own lives and decisions. Women's labour and knowledge are essential for achieving the goals of economic growth, environmental protection and improved quality of life. In Bangladesh, numerous initiatives have been developed to promote the full and equal participation of women in society (e.g., promoting education and employment opportunities), allowing them to reach their potentials and contribute to the development agenda. Access to education is significant for empowering women, as it opens up opportunities for them to participate in the labour market and secure financial independence and therefore contribute to economies and social development. Higher education and employment are two main pillars of women's empowerment as they can provide a greater autonomy, financial security and a better social standing [3–5]. In addition

to the better financial and social prospects, women's empowerment has been shown to be strongly associated with children's health outcomes [3–6] such as malnutrition [7–9], which is a major public health issue in developing countries like Bangladesh. Education is also an essential tool for preventing early marriage in women and promoting empowerment [10,11]. Early marriage is a widely prevalent issue across South Asia where girls are married off before the age of eighteen, sometimes against their will [12,13]. As a result, they are deprived of education and other opportunities that could help increase their socioeconomic standing. Education plays a major role in reducing the scope of early marriage, helps young girls understand their rights and empowers them to make informed decisions about their life and career [14]. Therefore, creating educational opportunities leads to more involvement from women in political decision-making processes and other leadership roles within society, which can help achieve gender equality and create the avenue for greater social stability [15].

Empowerment among women has also been found to be strongly associated with better decision-making autonomy, which can have significant implications for their health outcomes [16]. In the context of pregnancy and childbirth, for instance, women who are empowered are more likely to seek and receive timely and appropriate healthcare services, leading to better maternal and neonatal health outcomes [16]. Studies have shown that when women have greater decision-making power and control over their healthcare, they are more likely to use contraception, seek antenatal care, have a skilled attendant present during childbirth and seek timely treatment for complications if they arise [17–19]. This not only improves their own health and wellbeing but also has broader implications for the health and survival of their children, and the wellbeing of their families and communities. Therefore, efforts to empower women and promote their decision-making autonomy can lead to significant improvements in maternal and child health outcomes, and ultimately contribute to the achievement of local and global health goals.

Like in many low-middle-income countries, Bangladesh has also made appreciable progress in terms of women's empowerment during the last two decades, marked by higher school enrolment rates and labour market participation [20–22]. The growth of the garment and manufacturing industry has created job opportunities for a significant number of women, allowing them to improve their living conditions and provide better prospects for their children. Despite this impressive progress, many challenges remain in reaching true gender parity, including a gender gap in access to higher education and quality employment opportunities. The situation is particularly difficult for many married women as they are expected to be responsible primarily for the caregiving of the family [23–25]. Marriage is a significant part of many women's lives and can be an exciting and satisfying experience. However, it can also bring with it certain challenges that can make continuing to work or study difficult due to the traditional house-centred roles observed in patriarchal societies [26–28]. For many women, the decision to focus on their marriage and family obligations, rather than a career or studies, is an understandable but unfortunate outcome of the social norm. Currently, there is not enough research evidence regarding the barriers faced by young women in continuing work or studies post-marriage. This research is therefore aimed at exploring the factors associated with discontinuing work and studies after marriage among women aged 15–49 years. The findings can benefit women's empowerment programs and inform policy makers to develop evidence-based decisions.

## 2. Methods

### 2.1. Data Source

Data for this study were obtained from the latest Bangladesh Demographic and Health Survey (BDHS 2017–18), which is a nationally representative and cross-sectional survey that collects data on a wide range of demographic, healthcare and social factors. BDHS is conducted under the guidance of the National Institute of Population Research and Training (NIPORT) with technical assistance from ICF International through the DHS Program.

BDHS used a two-stage stratified sample of households, with 675 EAs (250 in urban areas and 425 in rural areas) selected in the first stage, and a systematic selection of an average of 30 households per Enumeration Area in the second stage. This resulted in a total of 20,250 residential households; however, interviews were conducted with 19,457 (20,376 ever-married women aged 15–49 years) households as the rest were unavailable during the survey. The sample population for the purpose of this study was limited to those who provided data on post-marriage study and work.

### 2.2. Study Variables

The outcome measures of the study included the post-marriage education and employment status of women. Both were categorized as "No", less than a year, 1–2 years, 3–4 years and 5 or more years. For the purpose of a regression analysis, the "Yes" category was created by combining less than a year, 1–2 years, 3–4 years and 5 or more years. The explanatory variables were selected based on their theoretical association with the outcome and included Respondent's current age; Religion (Islam/Hinduism/Buddhism/Christianity); Owns a mobile telephone (No/Yes); Access to electronic media (No/Yes); Travel decision maker (Respondent Alone, Respondent and Husband/Partner, Husband/Partner Alone, Other); Parity; Husband's highest year of education; Spousal age difference (0–4/5–9/10–14/>15); Sex of household head (Male/Female); Number of household members; Wealth index (Poorest/Poorer/Middle/Richer/Richest); Source of drinking water (Unimproved/Improved); Type of toilet facility (Unimproved/Improved); Division (Barisal/Chittagong/Dhaka/Khulna/Mymensingh/Rajshahi/Rangpur/Sylhet); and Type of place of residence (Urban/Rural).

### 2.3. Data Analysis

The study employed a descriptive analysis and binary logistic regression models to examine the proportions of the outcome measures (post-marriage education and employment) and their association with the explanatory variables. All analyses were conducted using Stata 16 using the *svy* command to account for the complex survey design. The first step was to calculate the percentage of the outcome variables using the descriptive analysis. Then, four binary logistic regression models were run. The first model included individual-label variables such as respondent's age, religion, access to mobile phone, access to electronic media, decision-making autonomy to travel and number of children. The second model additionally accounted for spouse-level variables such as husband's highest year of education and spousal age difference. The third model additionally accounted for and included household-level variables such as household head's sex, number of household members, household wealth index, household water source and type of toilet facilities. Finally, the fourth model added community-level variables such as division and residency type. The step-by-step addition of explanatory variables helped control for potential confounding variables and assess their impact on the relationship between the explanatory variables and the outcome variable. Finally, to ensure that the final model did not have any multicollinearity, the multicollinearity among the variables was examined using the variance inflation factor (VIF).

### 3. Results

Sociodemographic characteristics of the sample participants is available in the Appendix A (Table A1).

Figure 1 shows that a large majority of the participants did not continue studies (42.1%) and work (72.5%) after marriage. The percentage of participants who continued to work after marriage is comparatively higher than those who continued their studies. Only three percent of the participants studied for >5 years while about 29% worked for >5 years after marriage.

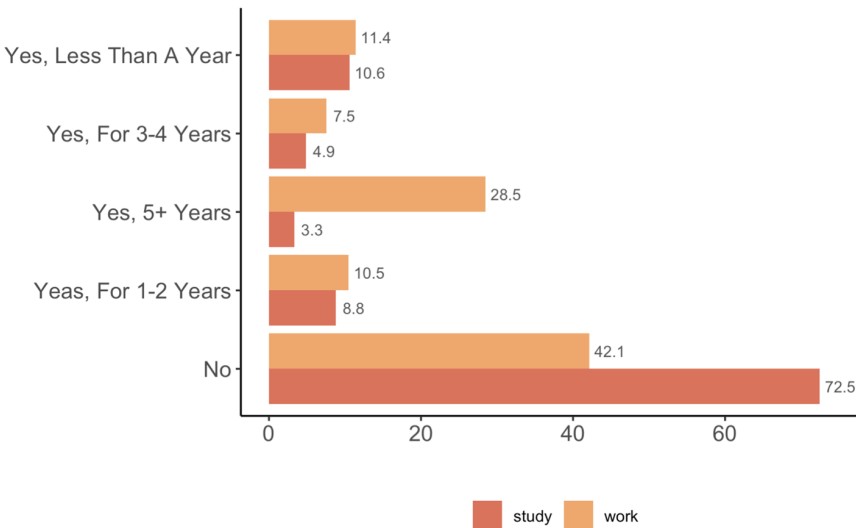

**Figure 1.** Percentage of women who continued to study and work after marriage.

Figure 2 shows the percentage of women who continued to study (A) and work (B) after marriage. Chattogram has the lowest percentage of women who continued to study after marriage (19.03%), followed by Sylhet (26.48%) and Rajshahi (24.90%). In contrast, Barisal has the highest percentage (31.92%), followed by Dhaka (30.49%) and Mymensingh (30.57%). Dhaka has also the highest percentage of women who continued to work after marriage (63.48%) followed by Chattogram (54.65%) and Sylhet (61.34%). In contrast, Barisal has the lowest percentage at 59.23%, followed by Mymensingh (57.25%) and Rangpur (56.43%).

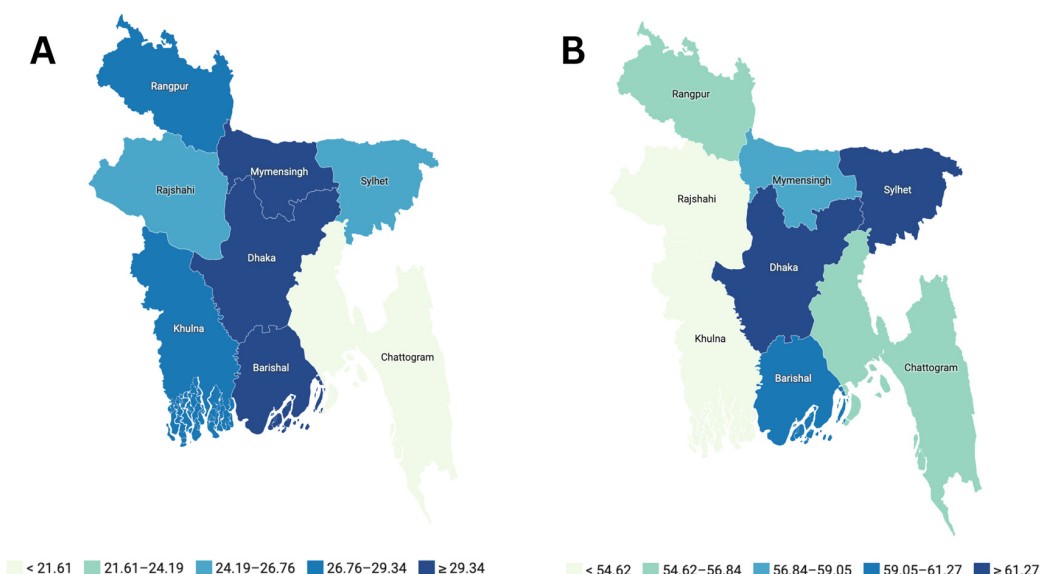

**Figure 2.** Percentage of women who continued to study (**A**) and work (**B**) after marriage in the seven divisions.

Sociodemographic characteristics of the sample population are presented in the Appendix A. Table 1 shows the sociodemographic factors associated with post-marriage studying among Bangladeshi women. As shown in the table, women who have a mobile phone are almost twice (OR = 1.89, 95% CI = 1.62, 2.19) as likely to continue studying after marriage compared to those who do not have a mobile phone. Having access to electronic media also showed a positive association with post-marriage studying (OR = 1.18, 95% CI = 1.01, 1.39). The number of children showed an inverse association with the con-

tinuation of studies after marriage (OR = 0.69, 95% CI = 0.65, 0.73). The husband's number of years of education showed a positive association with the continuation of studies after marriage (OR = 1.11, 95% CI = 1.08, 1.15). The odds ratio for the household head's sex (female) is 1.00 (95% CI = 0.84, 1.18), indicating that there is no significant association between household head's sex and continuing to study after marriage. A higher number of household members showed a slight increase in the odds of continuing to study after marriage (OR = 1.02, 95% CI = 1.00, 1.05).

**Table 1.** Factors associated with continuing to study after marriage among Bangladeshi women.

| | Model 1 | Model 2 | Model 3 | Model 4 |
|---|---|---|---|---|
| **Respondent's age** | 1.01 [1.00,1.02] | 1.01 [1.00,1.02] | 0.99 [0.98,1.00] | 0.99 [0.98,1.00] |
| **Religion (Islam)** | | | | |
| Hinduism | 0.84 [0.70,1.01] | 0.81 * [0.67,0.98] | 0.85 [0.70,1.03] | 0.86 [0.70,1.05] |
| Buddhism | 0.85 [0.35,2.08] | 0.87 [0.35,2.15] | 0.85 [0.34,2.11] | 1.38 [0.54,3.50] |
| Christianity | 2.56 [0.54,12.13] | 2.56 [0.53,12.36] | 2.35 [0.47,11.71] | 2.52 [0.50,12.60] |
| **Has mobile phone (No)** | | | | |
| Yes | 2.67 *** [2.33,3.07] | 2.43 *** [2.11,2.80] | 1.87 *** [1.62,2.17] | 1.89 *** [1.62,2.19] |
| **Access to electronic media (No)** | | | | |
| Yes | 1.85 *** [1.61,2.13] | 1.77 *** [1.54,2.05] | 1.18 * [1.01,1.39] | 1.18 * [1.01,1.39] |
| **Making decision to travel (Respondent Alone)** | | | | |
| Respondent and Husband/Partner | 1.18 [0.96,1.44] | 1.12 [0.91,1.38] | 1.09 [0.88,1.35] | 1.11 [0.89,1.37] |
| Husband/Partner Alone | 0.86 [0.68,1.09] | 0.85 [0.67,1.08] | 0.87 [0.68,1.11] | 0.90 [0.70,1.16] |
| Other | 1.12 [0.86,1.45] | 1.04 [0.80,1.36] | 0.94 [0.71,1.24] | 1.01 [0.76,1.33] |
| **Number of children** | 0.63 *** [0.60,0.67] | 0.65 *** [0.61,0.68] | 0.68 *** [0.64,0.72] | 0.69 *** [0.65,0.73] |
| **Husband's highest year of education** | | 1.15 *** [1.11,1.18] | 1.11 *** [1.08,1.15] | 1.11 *** [1.08,1.15] |
| **Spousal age difference (0–4)** | | | | |
| 5–9 | | 1.02 [0.88,1.18] | 0.98 [0.84,1.13] | 1.01 [0.87,1.17] |
| 10–14 | | 1.16 [0.99,1.36] | 1.04 [0.89,1.23] | 1.14 [0.96,1.34] |
| ≥15 | | 1.08 [0.87,1.34] | 0.97 [0.77,1.20] | 1.09 [0.87,1.37] |
| **Household head's sex (Male)** | | | | |
| Female | | | 0.96 [0.81,1.13] | 1.00 [0.84,1.18] |
| **Number of household members** | | | 1.01 [0.99,1.03] | 1.02 * [1.00,1.05] |

Table 1. *Cont*.

| | Model 1 | Model 2 | Model 3 | Model 4 |
|---|---|---|---|---|
| **Household wealth index (Lowest)** | | | | |
| Lower | | | 1.14 [0.85,1.53] | 1.27 [0.95,1.71] |
| Middle | | | 1.56 ** [1.18,2.06] | 1.81 *** [1.36,2.41] |
| Higher | | | 2.06 *** [1.55,2.73] | 2.41 *** [1.81,3.22] |
| Highest | | | 3.54 *** [2.66,4.73] | 4.31 *** [3.18,5.85] |
| **Household water source (Unimproved)** | | | | |
| Improved | | | 1.13 [0.87,1.48] | 1.05 [0.81,1.36] |
| **Type of toilet facilities (Unimproved)** | | | | |
| Improved | | | 1.34 ** [1.11,1.62] | 1.36 ** [1.12,1.65] |
| **Division (Barisal)** | | | | |
| Chittagong | | | | 0.29 *** [0.23,0.37] |
| Dhaka | | | | 0.45 *** [0.36,0.56] |
| Khulna | | | | 0.60 *** [0.49,0.75] |
| Mymensingh | | | | 0.80 [0.63,1.01] |
| Rajshahi | | | | 0.59 *** [0.47,0.73] |
| Rangpur | | | | 0.85 [0.68,1.06] |
| Sylhet | | | | 0.45 *** [0.34,0.61] |
| **Residency type (Urban)** | | | | |
| Rural | | | | 0.78 *** [0.68,0.88] |

Odds ratios with 95% confidence intervals in brackets. Level of significance: * $p < 0.05$, ** $p < 0.01$, *** $p < 0.001$. Model 1 = Respondent's age + Religion + Has mobile phone + Access to electronic media + Making decision to travel + Number of children. Model 2 = Model 1 + Husband's highest year of education + Spousal age difference. Model 3 = Model 2 + Household head's sex + Number of household members + Household wealth index + Household water source + Type of toilet facilities. Model 4 = Model 3 + Division + Residency type.

The odds of continuing to study after marriage also increased with a higher household wealth index: OR = 1.27 (95% CI = 0.95, 1.71). For a lower wealth index, OR = 1.81 (95% CI = 1.36, 2.41); for middle, OR = 2.41 (95% CI = 1.81, 3.22); and OR = 4.31 (95% CI = 3.18, 5.85) for the highest wealth index. Improved toilet facilities also showed a positive association with continuing to study after marriage (OR = 1.36, 95% CI = 1.12, 1.65). Compared to those in Barisal, women in Chittagong (OR = 0.29; 95% CI = 0.23, 0.37), Dhaka (OR = 0.45, 95% CI = 0.36, 0.56), Rajshahi (OR = 0.59; 95% CI = 0.47, 0.73) and Sylhet (OR = 0.45; 95% CI = 0.34, 0.61) are less likely to continue studying after marriage. Women living in rural areas (OR = 0.78, 95% CI = 0.68, 0.88) are less likely to continue studying after marriage compared to those living in urban areas.

Table 2 shows that compared with women who do not have a mobile phone, those who have one have higher odds (OR = 1.47, 95% CI = 1.06, 2.03) of working after marriage. Women whose spousal age difference is 15 years or more (compared with 0–4 years) showed a negative association (OR = 0.33, 95% CI = 0.19, 0.58) with continuing to work after marriage. Women from the Chittagong (OR = 0.53, 95% CI = 0.30, 0.96) division are less likely to continue working after marriage than women from the Barisal division.

**Table 2.** Factors associated with continuing to work after marriage among Bangladeshi women.

| | Model 1 | Model 2 | Model 3 | Model 4 |
|---|---|---|---|---|
| **Respondent's age** | 1.04 *** [1.02,1.05] | 1.03 ** [1.01,1.06] | 1.03 [1.00,1.05] | 1.03 [1.00,1.05] |
| **Religion (Islam)** | | | | |
| Hinduism | 1.06 [0.72,1.56] | 0.89 [0.58,1.38] | 0.95 [0.61,1.48] | 0.97 [0.62,1.53] |
| Buddhism | 2.13 [0.56,8.04] | 1.52 [0.39,5.88] | 1.76 [0.45,6.89] | 2.38 [0.58,9.76] |
| Christianity | 1.03 [0.17,6.36] | 0.65 [0.09,4.80] | 0.55 [0.07,4.07] | 0.53 [0.07,3.94] |
| **Has mobile phone (No)** | | | | |
| Yes | 1.69 *** [1.29,2.20] | 1.62 ** [1.19,2.20] | 1.50 * [1.09,2.07] | 1.47 * [1.06,2.03] |
| **Access to electronic media (No)** | | | | |
| Yes | 1.19 [0.90,1.58] | 1.22 [0.88,1.68] | 1.06 [0.73,1.53] | 1.07 [0.73,1.56] |
| **Making decision to travel (Respondent Alone)** | | | | |
| Respondent and Husband/Partner | 0.95 [0.62,1.45] | 0.94 [0.55,1.59] | 0.94 [0.55,1.61] | 0.92 [0.54,1.59] |
| Husband/Partner Alone | 0.92 [0.55,1.52] | 0.78 [0.42,1.44] | 0.81 [0.43,1.50] | 0.82 [0.44,1.52] |
| Other | 0.49 * [0.26,0.91] | 0.45 * [0.23,0.91] | 0.50 [0.25,1.01] | 0.49 * [0.24,1.00] |
| **Number of children** | 0.94 [0.84,1.04] | 0.91 [0.81,1.03] | 0.95 [0.84,1.08] | 0.95 [0.83,1.08] |
| **Husband's highest year of education** | | 1.02 [0.95,1.10] | 1.03 [0.95,1.11] | 1.03 [0.95,1.11] |
| **Spousal age difference (0–4)** | | | | |
| 5–9 | | 0.80 [0.59,1.08] | 0.81 [0.60,1.10] | 0.84 [0.62,1.14] |
| 10–14 | | 0.76 [0.51,1.13] | 0.79 [0.53,1.17] | 0.82 [0.55,1.24] |
| ≥15 | | 0.32 *** [0.18,0.56] | 0.32 *** [0.18,0.56] | 0.33 *** [0.19,0.58] |
| **Household head's sex (Male)** | | | | |
| Female | | | 1.06 [0.69,1.63] | 1.05 [0.68,1.62] |
| **Number of household members** | | | 0.95 [0.90,1.00] | 0.96 [0.91,1.01] |

**Table 2.** *Cont.*

| | Model 1 | Model 2 | Model 3 | Model 4 |
|---|---|---|---|---|
| **Household wealth index (Lowest)** | | | | |
| Lower | | | 1.04 [0.58,1.87] | 1.14 [0.63,2.07] |
| Middle | | | 1.17 [0.65,2.13] | 1.22 [0.67,2.25] |
| Higher | | | 1.49 [0.83,2.67] | 1.52 [0.82,2.81] |
| Highest | | | 1.47 [0.79,2.74] | 1.49 [0.77,2.91] |
| **Household water source (Unimproved)** | | | | |
| Improved | | | 1.00 [0.59,1.69] | 0.98 [0.58,1.66] |
| **Type of toilet facilities (Unimproved)** | | | | |
| Improved | | | 1.31 [0.86,2.00] | 1.30 [0.85,2.00] |
| **Division (Barisal)** | | | | |
| Chittagong | | | | 0.53 * [0.30,0.96] |
| Dhaka | | | | 0.83 [0.48,1.45] |
| Khulna | | | | 0.65 [0.35,1.20] |
| Mymensingh | | | | 0.88 [0.49,1.58] |
| Rajshahi | | | | 0.60 [0.33,1.10] |
| Rangpur | | | | 0.71 [0.39,1.29] |
| Sylhet | | | | 0.75 [0.40,1.42] |
| **Residency type (Urban)** | | | | |
| Rural | | | | 0.88 [0.64,1.21] |

Odds ratios with 95% confidence intervals in brackets. Level of significance: * $p < 0.05$, ** $p < 0.01$, *** $p < 0.001$. Model 1 = Respondent's age + Religion + Has mobile phone + Access to electronic media + Making decision to travel + Number of children. Model 2 = Model 1 + Husband's highest year of education + Spousal age difference. Model 3 = Model 2 + Household head's sex + Number of household members + Household wealth index + Household water source + Type of toilet facilities. Model 4 = Model 3 + Division + Residency type.

## 4. Discussion

Findings suggest that a large proportion of women discontinue their studies and work after marriage. There was also a considerable disparity in the proportion across the seven divisions of the country. About one fifth of the participants in Chattogram reported continuing studies after marriage compared to about one-third in Barisal. However, Barisal has the lowest percentage of women who continued to work after marriage. These statistics might be reflective of the regional differences in social norms that expect women to prioritize household responsibilities and caregiving roles over their own education, careers or aspirations after marriage. These traditional gender roles and expectations that potentially curb women's empowerment should be explored through qualitative studies. The results

highlight the need for policy interventions to promote women's education and participation in the workforce even after marriage.

The results suggest that access to technology and resources, such as mobile phones and electronic media, is positively associated with the likelihood of women continuing their education after marriage. This highlights the importance of improving women's access to technology and resources, particularly in rural areas where access to such resources may be limited. Electronic media may also provide educational opportunities and resources that were previously unavailable or difficult to access for women, particularly in rural areas [29,30]. The negative association between the number of children and continuing education after marriage underscores the need for policies and programs that support women's reproductive rights and access to family planning. Women with more children may have more domestic and caregiving responsibilities, which can limit the amount of time and resources they have available to devote to their education [31,32]. Policies that support maternity leave and affordable childcare can help to alleviate the challenges of balancing education and caregiving responsibilities for married women or new mothers. The positive association between the husband's education level and women's continuation of education after marriage emphasizes the importance of improving access to education for both men and women. Husbands with higher educational levels are more likely to have a positive attitude towards women's education and are more supportive of their wives continuing their education after marriage [33]. Results also suggest that the household size plays a minor role in women's education after marriage. This might be because a larger household may imply a stronger familial support system, both emotionally and financially, in fulfilling domestic responsibilities that can help women to continue their education. Alternatively, it is also possible that larger households have a greater division of labour, which can allow women to have more spare time and freedom to pursue their studies or external work.

The findings suggest that access to resources such as mobile phones, education and wealth is positively associated with the likelihood of continuing to study after marriage in Bangladesh. On the other hand, factors such as having children, living in rural areas and residing in certain divisions have a negative association with continuing education. Similarly, women with a mobile phone are more likely to continue working after marriage, while those with larger spousal age differences and those living in the Chittagong division are less likely to do so. Qualitative studies are required in this line of research to inform policy targeting the promotion of socioeconomic empowerment opportunities among women after marriage. The results highlight the importance of addressing socioeconomic factors to promote women's empowerment and gender equality in Bangladesh. Policies and programs that improve women's access to technology and resources, particularly in rural areas, can positively impact women's education after marriage. Additionally, policies that support women's reproductive rights and access to family planning can help to reduce the burden of caregiving responsibilities and enable women to pursue their educational goals.

To the best of our knowledge, this is the first study to explore the factors associated with post-marriage study and work among women in Bangladesh. The sample size was relatively large and the analysis included a wide range of individual-, household- and community-level variables. These findings capture the socioeconomic context in which women live and provide insights into barriers that women face in pursuing higher education. The findings from the study can inform policy interventions by identifying which groups of women are most disadvantaged and may need greater support. There are several limitations of the study that must be taken into account when interpreting the findings. Firstly, this study focuses specifically on measuring changes in access to education and employment opportunities among women after marriage, and does not directly measure changes in women's ability to make choices. While these measurable changes suggest a decrease in certain aspects of empowerment, the study is limited in that it does not capture any changes in women's capacity to make choices. Declines in education or employment do not necessarily mean women have less agency overall. Given the secondary nature of the data,

we could not consider the broader sociocultural and institutional factors like social norms, gender roles, access to childcare and workplace policies, which may have a strong impact on post-marriage study and work. Similarly, the study also lacks an in-depth qualitative insight into women's experiences, challenges, motivations and thought processes, which can provide a richer understanding of their empowerment issues. Finally, the study cannot determine the causal relationships due to the cross-sectional nature of the data.

## 5. Conclusions

In conclusion, the study found that a significant proportion of women in Bangladesh do not continue their studies or work after marriage, but some factors such as access to a mobile phone, the husband's education level, household wealth and improved toilet facilities were found to be positively associated with continuing education after marriage. On the other hand, the number of children, living in rural areas and residing in certain divisions were negatively associated with continuing education. Similarly, women with a mobile phone were found to be more likely to continue working after marriage, while larger spousal age differences and residing in the Chittagong division were negatively associated with continuing work after marriage. These findings highlight the importance of addressing structural and social factors to empower women and enable them to make decisions that can positively impact their education and work opportunities.

**Author Contributions:** Conceptualization, B.G.; methodology, B.G.; software, B.G.; validation, B.G.; I.H. and A.A.M.; formal analysis, B.G.; investigation, B.G., I.H. and A.A.M.; resources, B.G.; data curation, B.G.; writing—original draft preparation, B.G.; writing—review and editing, B.G., I.H. and A.A.M.; visualization, B.G., I.H. and A.A.M.; supervision, B.G., I.H. and A.A.M.; project administration, B.G.; I.H. and A.A.M; funding acquisition, B.G. All authors have read and agreed to the published version of the manuscript.

**Funding:** This research received no external funding.

**Institutional Review Board Statement:** Ethical review and approval were waived for this study as the data were secondary.

**Informed Consent Statement:** Informed consent was obtained from all subjects involved in the study.

**Data Availability Statement:** Data are available through the DHS website.

**Conflicts of Interest:** The authors declare no conflict of interest.

## Appendix A

**Table A1.** Sample characteristics, % (95% CI).

| Variables | Total (8041) |
| --- | --- |
| Respondent's current age, mean (SD) | 28.4 (8.4) |
| Religion (Islam) | 90.9 (90.3; 91.6) |
| Religion (Hinduism) | 8.6 (8.0; 9.2) |
| Religion (Buddhism) | 0.3 (0.2; 0.4) |
| Religion (Christianity) | 0.1 (0.0; 0.2) |
| Owns a mobile telephone (No) | 28.2 (27.3; 29.2) |
| Owns a mobile telephone (Yes) | 71.8 (70.8; 72.7) |
| Access to electronic media (No) | 25.4 (24.4; 26.3) |
| Access to electronic media (Yes) | 74.6 (73.7; 75.6) |
| Travel decision maker (Respondent Alone) | 7.9 (7.3; 8.5) |
| (Respondent and Husband/Partner) | 64.5 (63.4; 65.5) |
| (Husband/Partner Alone) | 18.0 (17.2; 18.9) |
| (Other) | 9.7 (9.0; 10.3) |
| Parity, mean (SD) | 2.2 (1.3) |

**Table A1.** *Cont.*

| Variables | Total (8041) |
|---|---|
| Husband's highest year of education, mean (SD) | 3.4 (1.8) |
| Age difference (0–4) | 21.7 (20.8; 22.6) |
| Age difference (5–9) | 43.6 (42.5; 44.7) |
| Age difference (10–14) | 25.3 (24.3; 26.3) |
| Age difference (>15) | 9.5 (8.8; 10.1) |
| Sex of household head (Male) | 85.7 (84.9; 86.5) |
| Sex of household head (Female) | 14.3 (13.5; 15.1) |
| Number of household members, mean (SD) | 5.4 (2.6) |
| Wealth index combined (Poorest) | 11.4 (10.7; 12.1) |
| Wealth index combined (Poorer) | 14.7 (13.9; 15.5) |
| Wealth index combined (Middle) | 20.1 (19.2; 21.0) |
| Wealth index combined (Richer) | 23.0 (22.1; 23.9) |
| Wealth index combined (Richest) | 30.8 (29.8; 31.8) |
| Source of drinking water (Unimproved) | 12.6 (11.9; 13.3) |
| Source of drinking water (Improved) | 87.4 (86.7; 88.1) |
| Type of toilet facility (Unimproved) | 29.3 (28.4; 30.3) |
| Type of toilet facility (Improved) | 70.7 (69.7; 71.6) |
| Division (Barisal) | 12.0 (11.3; 12.7) |
| Division (Chittagong) | 13.3 (12.6; 14.1) |
| Division (Dhaka) | 15.0 (14.2; 15.8) |
| Division (Khulna) | 16.3 (15.5; 17.1) |
| Division (Mymensingh) | 9.6 (9.0; 10.2) |
| Division (Rajshahi) | 15.1 (14.3; 15.9) |
| Division (Rangpur) | 13.0 (12.3; 13.7) |
| Division (Sylhet) | 5.7 (5.2; 6.2) |
| Type of place of residence (Urban) | 39.1 (38.0; 40.2) |
| Type of place of residence (Rural) | 60.9 (59.8; 62.0) |

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
