# Peer review of "Factors Influencing Post-Marriage Education and Employment among Bangladeshi Women: A Cross-Sectional Analysis"

_women, doi:10.3390/women3030030_

Round 1
Reviewer 1 Report
This study examines factors associated with education and employment following marriage among Bangladeshi women. The paper tackles an important problem, but the following issues need attention.
1. The literature review is quite lacking. There is a lot of research on women's barriers to education and employment after marriage. While such research is not focused on Bangladeshi women (see work in other countries like the U.S.), it can provide a foundation for this study. Such research needs to be reviewed.
2. There is also some research on how religion influences these types of life opportunities. There were religion measures featured in the survey, so the relevant research should be reviewed.
3. More attention is needed concerning the cultural ideologies that sustain these gender differences. In much of the West, the ideology of separate spheres was the dominant framework that shaped gender stratification by restricting women to the "private sphere." Does that ideology apply to this case? Regardless, such work should be reviewed and an effort to explain how these outcomes occur should be offered. Michael Kimmel argues that ideologies of gender difference (essentialism, etc.) are often used to justify and sustain gender dominance (inequality among men and women).
4. All abbreviations should be spelled out before the abbreviations are used. EAs is one example.
5. What are the sample limitations beyond the restriction of the study sample to those who are married and answered the employment and education questions? Was oversampling used and, if so, were weights applied?
6. At some points, causal order is presumed. Does having a phone actually facilitate education and employment or is such a device seen as a necessary communication tool for women who are already in the public sphere? Causal inferences should be made with extreme care.
A good study, but one that could be improved in various ways.
This paper could benefit from a careful proofreading. The issues are even evident in the abstract and the very first sentence of the manuscript, where "women empowerment" should be "women's empowerment." Editing by a highly proficient or native English speaker is recommended.
Author Response
- The literature review is quite lacking. There is a lot of research on women's barriers to education and employment after marriage. While such research is not focused on Bangladeshi women (see work in other countries like the U.S.), it can provide a foundation for this study. Such research needs to be reviewed.
Response: Thanks for this suggestion. We have added some more content education in light of the existing literature. Please let us know if this fine.
- There is also some research on how religion influences these types of life opportunities. There were religion measures featured in the survey, so the relevant research should be reviewed.
Response: Thanks indeed for this suggestion. Religion didn’t not show any significant association with the outcome variables, which is why we didn’t spend extra space on it. Please advise on what can we add to address this.
- More attention is needed concerning the cultural ideologies that sustain these gender differences. In much of the West, the ideology of separate spheres was the dominant framework that shaped gender stratification by restricting women to the "private sphere." Does that ideology apply to this case? Regardless, such work should be reviewed and an effort to explain how these outcomes occur should be offered. Michael Kimmel argues that ideologies of gender difference (essentialism, etc.) are often used to justify and sustain gender dominance (inequality among men and women).
Response: Thanks for the suggestion. The scope of the questions is beyond our expertise, and probably of the analysis as well. Our goal is to interpret and describe what the data is telling, and as quantitative researchers, our ability to interpretation is limited to what the result is portraying. We’d be really grateful if you could provide an idea on which we could build the suggested discussion!
- All abbreviations should be spelled out before the abbreviations are used. EAs is one example.
Response: EA was now spelled out. Enumeration Areas
- What are the sample limitations beyond the restriction of the study sample to those who are married and answered the employment and education questions? Was oversampling used and, if so, were weights applied?
Response: As far as we are concerned, DHS surveys do not use oversampling. We used the svy command in Stata to account for sampling weights, and mentioned this in the analysis as well.
- At some points, causal order is presumed. Does having a phone actually facilitate education and employment or is such a device seen as a necessary communication tool for women who are already in the public sphere? Causal inferences should be made with extreme care.
Response: Thanks for this suggestion. We have modified the text to avoid making cany causal inference.
A good study, but one that could be improved in various ways.
Response: Thanks indeed for your valuable comments. It helped us improve the quality of the writing.
Reviewer 2 Report
The paper has got a very suggestive title and a promissing idea about a very relevant topic such as Post-Marriage Education and Employment
among Bangladeshi Women. I have some questions, comments and suggestions for the authors:
Abstract: there is no mention to the aims/hypotheses in the abstract, only about the analysis. A failure in the abstract is to present same background ("many women fail to continue their studies or work after marriage which can significantly reduce their empower-ment potential, especially in countries with stark gender inequality such as in Bangladesh") than conclusion ("The study indicates that a large proportion of Bangladeshi
women don't continue their education or work after marriage"). Please, review and add relevant information.
- Aims of the study are not clearly presented, there is only one aim but there are no operative objectives in the study background or the Methods part.
-About the data source, if it's possible to include the validity & feasibility index of the survey, it would increase the confidability. Also, how had the participants expressed their opinions in the survey.
- Results are wonderful and very well presented, thanks a lot. Anyway, data in text are same than data in table, so it´would be more feasible for the reader to find in the text only significative data. I have one doubt about the Results, the best part of the paper: why there are no results about muslim religion? It's a very big lack, most of the sample it's not analyzed...
Discussion.
- To support your discussion, it´s necessary to make reference to the differences between religions (there is no mention in the text). The comment about "cultural and social factors" is not enough to explain the differences. Also, at the beginning and in the end of your discusion is necessary to include references about previous research on the topic.
- Maybe it would be interesting to include some practical proposals in Conclusion part
Author Response
The paper has got a very suggestive title and a promissing idea about a very relevant topic such as Post-Marriage Education and Employment among Bangladeshi Women. I have some questions, comments and suggestions for the authors:
Abstract: there is no mention to the aims/hypotheses in the abstract, only about the analysis. A failure in the abstract is to present same background ("many women fail to continue their studies or work after marriage which can significantly reduce their empower-ment potential, especially in countries with stark gender inequality such as in Bangladesh") than conclusion ("The study indicates that a large proportion of Bangladeshi women don't continue their education or work after marriage"). Please, review and add relevant information.
- Aims of the study are not clearly presented, there is only one aim but there are no operative objectives in the study background or the Methods part.
Response: Response: Thanks for the suggestion. We have now added the objectives in the abstract.
-About the data source, if it's possible to include the validity & feasibility index of the survey, it would increase the confidability. Also, how had the participants expressed their opinions in the survey.
-Response: We are not sure if DHS surveys provide information regarding ‘validity & feasibility index’. We have explored the survey methods but didn’t find any information. Participants expressed their opinions by answering the structured questionnaires.
Results are wonderful and very well presented, thanks a lot. Anyway, data in text are same than data in table, so it´would be more feasible for the reader to find in the text only significative data.
Response: Thanks for this comment. We have presented the significant results only.
I have one doubt about the Results, the best part of the paper: why there are no results about muslim religion? It's a very big lack, most of the sample it's not analyzed.
Response: Thanks for this comment. As shown in the appendix file, Muslim religion accounted for >90% of the participants. In the regression analysis, Muslim religion was used as the reference category. However, the association between religion and the outcomes variables was not significant and hence was not discussed.
Discussion.
- To support your discussion, it´s necessary to make reference to the differences between religions (there is no mention in the text). The comment about "cultural and social factors" is not enough to explain the differences. Also, at the beginning and in the end of your discusion is necessary to include references about previous research on the topic.
Response: Thanks for this comment. We agree that using the term ‘cultural factors’ is not appropriate in this context and hence have removed it from the statement. The lack of sociocultural factors in the analysis was mentioned in the limitations. As mentioned in the earlier response, the association between religion and the outcomes variables was not significant and hence was not discussed.
- Maybe it would be interesting to include some practical proposals in Conclusion part
Response:Thanks for this comment. We have added some policy recommendations in the conclusion.
Round 2
Reviewer 1 Report
I commend the authors on a sound revision. Well done.
Author Response
Thanks indeed for your valuable time to review our paper!
Reviewer 2 Report
I really appreciate the efforts of the authors and the inclusions of the suggestions. The paper now is ready for publication, in my opinion.
Author Response

(The authors gave the same response as above.)
